# Host biomarkers and combinatorial scores for the detection of serious and invasive bacterial infection in pediatric patients with fever without source

**Laurence Lacroix**[1,2]*, **Sebastien Papis**[3], **Chiara Mardegan**[3], **Fanny Luterbacher**[3], **Arnaud L'Huillier**[2,4], **Cyril Sahyoun**[1,2], **Kristina Keitel**[5], **Niv Mastboim**[6], **Liat Etshtein**[6], **Liran Shani**[6], **Einav Simon**[6], **Eran Barash**[6], **Roy Navon**[6], **Tanya M. Gottlieb**[6], **Kfir Oved**[6], **Eran Eden**[6], **Christophe Combescure**[7], **Annick Galetto-Lacour**[1,2‡], **Alain Gervaix**[1,2‡]

1 Pediatric Emergency Department, Children's Hospital, University Hospitals of Geneva, Geneva, Switzerland, 2 Faculty of Medicine, University of Geneva, Geneva, Switzerland, 3 Department of General Pediatrics, Children's Hospital, University Hospitals of Geneva, Geneva, Switzerland, 4 Department of Pediatric Infectious Diseases, Children's Hospital, University Hospitals of Geneva, Geneva, Switzerland, 5 Pediatric Emergency Department, Inselspital, Bern, Switzerland, 6 MeMed, Tirat Carmel, Israel, 7 Department of Clinical Epidemiology, University Hospitals of Geneva, Geneva, Switzerland

‡ AGL and AG are contributed equally to this work and joint senior authors.
* laurence.lacroix@hcuge.ch

**Data Availability Statement:** The data underlying the results presented in the study are available from https://doi.org/10.5061/dryad.gxd2547s8.

## Abstract

### Background

Improved tools are required to detect bacterial infection in children with fever without source (FWS), especially when younger than 3 years old. The aim of the present study was to investigate the diagnostic accuracy of a host signature combining for the first time two viral-induced biomarkers, tumor necrosis factor-related apoptosis-inducing ligand (TRAIL) and interferon γ-induced protein-10 (IP-10), with a bacterial-induced one, C-reactive protein (CRP), to reliably predict bacterial infection in children with fever without source (FWS) and to compare its performance to routine individual biomarkers (CRP, procalcitonin (PCT), white blood cell and absolute neutrophil counts, TRAIL, and IP-10) and to the Labscore.

### Methods

This was a prospective diagnostic accuracy study conducted in a single tertiary center in children aged less than 3 years old presenting with FWS. Reference standard etiology (bacterial or viral) was assigned by a panel of three independent experts. Diagnostic accuracy (AUC, sensitivity, specificity) of host individual biomarkers and combinatorial scores was evaluated in comparison to reference standard outcomes (expert panel adjudication and microbiological diagnosis).

### Results

241 patients were included. 68 of them (28%) were diagnosed with a bacterial infection and 5 (2%) with invasive bacterial infection (IBI). Labscore, ImmunoXpert, and CRP attained the

**Funding:** This was an investigator-driven study. Alain Gervaix received financial support from MeMed to support the study. The funder provided support in the form of salaries for authors NM, LE, LS, ES, EB, RN, TG, KO, EE. The specific roles of these authors are articulated in the 'author contributions' section. The funders and study sponsors had no role in the design and conduct of the study, and in interpretation of clinical data. The first version of the manuscript was written by Laurence Lacroix. The funder (MeMed) participated in data collection and analysis (ImmunoXpert assays) and in reviewing the manuscript. Annick Galetto-Lacour and Arnaud L'Huillier received funding to support the study from the Gertrude von Meissner Foundation, the Ernst and Lucie Schmidheiny Foundation and the University Hospitals of Geneva's Research and Development Project Grant. There was no additional external funding received for this study.

**Competing interests:** The financial support received by AG from MeMed as well as the commercial affiliation of authors NM, LE, LS, ES, EB, RN, TG, KO, EE with MeMed does not alter our adherence to all PLOS ONE policies on sharing data and materials. Other authors had no relationship relating to employment, consultancy, patents, products in development, marketed products, etc. with any of the funding organizations.

highest AUC values for the detection of bacterial infection, respectively 0.854 (0.804–0.905), 0.827 (0.764–0.890), and 0.807 (0.744–0.869). Labscore and ImmunoXpert outperformed the other single biomarkers with higher sensitivity and/or specificity and showed comparable performance to one another although slightly reduced sensitivity in children < 90 days of age.

## Conclusion

Labscore and ImmunoXpert demonstrate high diagnostic accuracy for safely discriminating bacterial infection in children with FWS aged under and over 90 days, supporting their adoption in the assessment of febrile patients.

## Introduction

Fever without source (FWS) is a common presenting symptom in children less than 3 years old [1]. It is defined as fever 38.0˚C/100.4˚F and above for less than 7 days with no underlying cause despite a thorough history and physical exam [2, 3]. Being able to differentiate serious bacterial infections (SBI) or invasive bacterial infections (IBI) necessitating immediate antibiotic treatment from focal bacterial infections or viral infections remains crucial although often challenging, because of lack of specificity in symptoms and signs [4, 5]. Diagnosis is therefore frequently supported by biological exams, such as determination of C-reactive protein (CRP) and procalcitonin (PCT) values. The Labscore takes into account three variables which are independently associated with SBI, weighed according to the odds ratio in the univariate analysis in the original derivation study [6]: 1) PCT ($\geq$2 ng/mL: 4 points, 0.5–2.0 ng/mL: 2 points, <0.5 ng/mL: 0 point) 2) CRP ($\geq$100 mg/L: 4 points, 40–99 mg/L: 2 points, < 40 mg/L: 0 point) and 3) urine dipstick (positive, ie, positive leukocyte esterase and/or positive nitrate: 1 point, negative: 0 point). Consequently, Labscore values range from 0 to 9 points. A cutoff point > = 3 was identified as the best Labscore value for SBI prediction, even when applied to a large external cohorts of children in the same age range [7, 8].

Whereas most biomarkers or combinations of biomarkers are responsive to bacterial infections, a new assay (ImmunoXpert) intended for children aged 90 days and more and adults has been recently developed. It combines for the first time two viral-induced biomarkers, tumor necrosis factor-related apoptosis-inducing ligand (TRAIL) and interferon γ-induced protein-10 (IP-10), with a bacterial-induced one, C-reactive protein (CRP) [9]. It displays three possible outcomes: (1) Viral infection (or non-bacterial etiology): ImmunoXpert score < 35; (2) Equivocal: 35 $\leq$ ImmunoXpert score $\leq$ 65; and (3) Bacterial infection (including mixed bacterial and viral co-infection): ImmunoXpert score > 65. This signature has already shown high diagnostic accuracy in a heterogeneous study population of febrile inpatients and emergency department arrivals, both children and adults, presenting with diverse clinical syndromes and pathogens [9–13]. Biomarkers or combination of biomarkers including the recently described ImmunoXpert have been frequently studied for their clinical utility to accurately guide antibiotic treatment in patients with acute respiratory tract infections [14].

The potential applicability of the TRAIL/IP-10/CRP assay and its comparison with other common biomarkers and the Labscore has never been studied prospectively in a population of children with FWS. Moreover, limited data were available in the fragile subpopulation of patients aged under 90 days in whom the need is serious for improved diagnostic tools to drive decision-making [15, 16]. The objective of the present study was to assess the diagnostic

accuracy of various individual and combined bacterial- and viral-specific biomarkers, notably the ImmunoXpert and the Labscore, in children and infants under and over 90 days old presenting with FWS.

## Material and methods

### Study design

Prospective diagnostic accuracy study in children with FWS aged less than 3 years old (consecutive sample), performed in the pediatric emergency department (PED) of a single Swiss tertiary center from November 2015 to January 2018, using expert panel adjudication and microbiological diagnosis as diagnostic reference standards.

**Primary objective.**   To prospectively investigate the diagnostic accuracy of a host signature (TRAIL, IP-10, and CRP) to reliably predict bacterial infections and to compare its performance to routine individual biomarkers (CRP, PCT, WBC, ANC, TRAIL, and IP-10) and to the Labscore.

**Secondary objectives.**   To compare the diagnostic accuracy of the ImmunoXpert, the Labscore, and individual biomarkers to reliably predict IBI.

### Participants

Patients aged less than 3 years old presenting to the PED with FWS after a thorough history and physical exam were eligible. Exclusion criteria were comorbidities predisposing to infections such as cancer, primary or secondary immunodeficiency, and iatrogenic immunosuppression.

### Study procedure

After informed consent was obtained, patients' characteristics were recorded (age, sex, delay between fever onset and presentation, maximum temperature before presentation). Patients were assessed with white blood cell count (WBC), absolute neutrophil count (ANC), band count, CRP, PCT, urinary dipstick and culture, and blood culture. Additional testing (CSF culture, chest X-ray, synovial fluid or stool culture etc.) was optional. An additional 0.6 mL sample was drawn, dedicated to the determination of CRP, TRAIL and IP-10 and the ImmunoXpert score. One of the objectives in our study was to assess the differential expression of biomarkers and combinational scores between patient with FWS and children with no fever and no sign for any current infection. Facing the ethical challenge of performing blood puncture in healthy control children with no fever and no current infection, relevant biomarkers from control patients were tested on preexisting blood samples. We did not find healthy pediatric control samples in Switzerland, but an existing healthy Canadian cohort could meet this specific objective. A group of 50 healthy Canadian children from trauma and dental clinics served as control patients. The full study protocol is available as online supplemental material (S1 File). The study was approved by the Institutional Ethics Committee (Cantonal Committee on Ethics in Scientific Research of Geneva, CCER), adheres to STARD 2015 guidelines (S1 Table) [17], and is registered under ClinicalTrials.gov (NCT03224026). No investigation was performed before signature of a written informed consent. The use of sera from control patients was approved by Toronto's Hospital for Sick Children Ethics Committee.

### Reference standard

Reference standard was based on expert panel adjudication, which is a common approach to assigning a final diagnosis in fields where gold standard is lacking, specifically in patients with

FWS [18, 19]. On termination of the recruitment phase, the experts (one senior pediatric infectious-disease specialist and two senior pediatric emergency physicians) independently performed an initial etiologic classification (viral, bacterial, or indeterminate). Then, they were asked to independently classify patients with indeterminate diagnosis into a dichotomous outcome (viral or bacterial infection), as recommended in most studies using this kind of reference standard [18]. A "majority" reference standard was established: the reference standard etiology (viral or bacterial infection) was the etiology label assigned by minimum 2 of the 3 experts.

Other definitions of reference standard were also investigated to assess the robustness of the findings. First microbiological diagnosis was established: (1) bacteriologically proven SBI or IBI or (2) no proof of bacterial infection according to predefined criteria. SBI was defined as isolation of a bacterial pathogen from any urine, synovial fluid, bone, or stool specimen [18]. Microbiological urinary tract infection (UTI) was defined according to the AAP Subcommittee on Urinary Tract Infection and Steering Committee on Quality Improvement and Management [20], on a urine specimen obtained through urinary tract catheterization or clean catch of mid-stream urine: (1) patients > 60 days old: abnormal urinalysis defined by the presence of positive leukocyte esterase, nitrite or pyuria (>5 white blood cells (WBCs) per high-power field) and culture growth of at least 50'000 colony-forming units (cfu) per mL of a uropathogen and (2) patients ≤ 60 days old (as recently described after adaptation from the American Academy of Pediatrics practice parameters [21]): growth of 50'000 cfu/mL or more of a uropathogen, or growth of 10'000 cfu/mL or more of a single uropathogen in association with an abnormal urinalysis. IBI was defined as isolation of a compatible bacterial pathogen in blood or cerebrospinal fluid culture. Second, a "unanimous" reference standard was considered: the reference standard etiology (viral or bacterial) was determined only if the same etiology label was assigned by all 3 experts. When analyzing this reference standard, children with an indeterminate etiology or showing equivocal ImmunoXpert score (35 ≤ ImmunoXpert score ≤ 65) were excluded from the subgroup analysis.

## Blinding procedure

Experts had access to full medical records but were blinded to the diagnosis of their peers and to TRAIL, IP-10 and ImmunoXpert results; CRP, PCT and urinary dipstick data were available to the experts. The ImmunoXpert test was performed on anonymized samples, and performers/readers of the index tests had no access to clinical information and reference standard results. The index test and panel expert reference standard outcomes were locked prior to unblinding at the end of the recruitment phase.

## Sample size

The sample size of this ancillary study was driven by a main study on viremia in patients with FWS, in which a sample of 50 viremic patients was planned [22]. To reach this target, assuming a 10–20% prevalence of viremia, the enrolment of 400 patients was anticipated. A power calculation for the ancillary study was conducted to determine the detectable difference in area under the ROC curve (of any biomarker compared with the ImmunoXpert) with a power of 80% under various sample sizes (0.08 detectable difference with a sample size of 100 to 0.10 with a sample size of 264 patients). The enrolment of patients terminated when the number of patients with viremia corresponded to the target. The power to detect a difference in areas under ROC curve was acceptable for the evaluation of biomarker accuracy.

Concerning power calculation, we assumed (1) a two-sided risk alpha of 0.05, (2) a correlation between ImmunoXpert and the other marker of 0.5 both in patients with a bacterial

infection and in those with a viral infection and (3) a viral infection rate 3 times higher than that of bacterial infection. Using the method of sample size calculation for comparison of paired binormal ROC curves proposed by Obuchowski et al [23], the needed sample size was 264 patients (66 with a bacterial infection and 198 with a viral infection).

### Statistical analysis

Markers were described in patients with bacterial and viral infections by median and inter-quartile intervals. Mann-Whitney test was used to compare these groups. Diagnostic performances of biological scores and markers in identifying bacterial infections were assessed by non-parametric ROC curves. The areas under the ROC curves were assessed and compared using the non-parametric approach proposed by Delong et al [24]. This approach accounted for paired data when two compared markers were measured in the same person. Sensitivities and specificities were reported with the Clopper-Pearson exact 95% confidence intervals. The two-sided risk alpha was 0.05 in all comparisons. Software used for statistical analyses was R version 3.5.2 (R Core Team (2018). R: A language and environment for statistical computing. R Foundation for Statistical Computing, Vienna, Austria. URL https://www.R-project.org/).

## Results

### Characteristics of the study population

Overall, 58120 patients attended our PED during the study period. Among 313 potentially eligible participants, 72 were excluded and 241 were eligible for testing. Reasons for exclusion, stratification by age groups and diagnostic classification by the panel experts are indicated in Fig 1.

Table 1 reports descriptive statistics for the main demographic and baseline epidemiologic variables. Two-thirds of patients were less than 90 days old (n = 155, 64%). They were more frequently male, showed a lower maximal temperature before presentation and presented more frequently in the first 12 hours from fever onset than older patients. The most frequent bacterial infection was urinary tract infection (UTI) (n = 56, 82%) (Table 1). Bacterial prevalence was 29% and 27% among children aged under and ≥ 90 days respectively.

Fifty healthy patients were included as a control group. Controls were older than patients in the study group, with a median age of 18.1 months (IQR: 13.0 to 25.6) versus 2 months (IQR: 1.0 to 4.9) (p<0.0001).

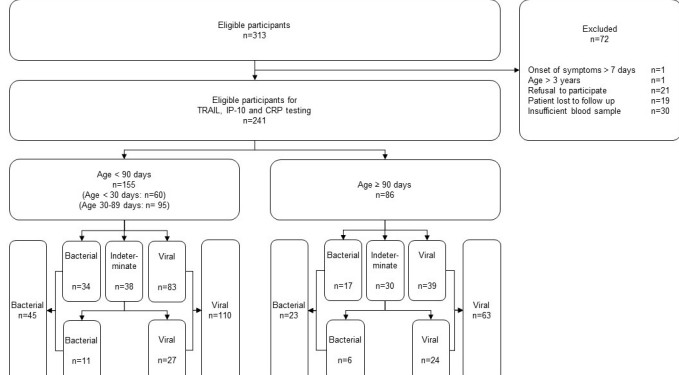

**Fig 1. Flowchart of the study.** Study flowchart indicating eligible patients, reasons for exclusion, stratification according to age groups and etiology diagnosis assigned by the expert panel.

**Table 1. Baseline and diagnostic characteristics of study population.**

| Patient characteristics | Overall | < 90 days | ≥ 90 days | p |
|---|---|---|---|---|
| **Total, n** | 241 | 155 | 86 | |
| **Sex** | | | | |
| Female, n (%) | 93 (39) | 51 (33) | 42 (49) | 0.022 |
| Male, n (%) | 148 (61) | 104 (67) | 44 (51) | |
| **Delay between fever onset and presentation < 12 h, n (%)** | 105 (44) | 90 (58) | 15 (17) | <0.0001 |
| **Maximum temperature before presentation [˚C], mean (SD)** | 39.1 (0.8) | 38.8 (0.5) | 39.7 (0.7) | <0.0001 |
| **Diagnosis according to expert panel (majority cohort)** | | | | |
| Viral infection, n (%) | 173 (72) | 110 (71) | 63 (73) | 0.819 |
| Bacterial infection, n (%) | 68 (28) | 45 (29) | 23 (27) | |
| Urinary tract infection, n[a] | 56 (82) | 41 (91) | 15 (65) | |
| Suspected bacterial infection without any source found, n[a] | 4 (6) | 2 (4) | 2 (9) | |
| Meningitis, n[a] | 3 (4) | 1 (2) | 3 (13) | |
| Bacteremia, n[a] | 2 (3) | 1 (2) | 0 (0) | |
| Pneumonia, n[a] | 1 (1) | 0 (0) | 1 (4) | |
| Other, n[a] | 2 (3) | 0 (0) | 2[b] (9) | |
| **Microbiological diagnosis, n (%)** | | | | |
| Bacterial infection (SBI and IBI) | 41 (17) | 29 (19) | 12 (14) | 0.446 |
| IBI | 5 (2) | 2[c] (1) | 3[d] (3) | 0.352 |

[a] % within bacterial infection

[b] Perforated appendicitis (n = 1), acute otitis media (n = 1)

[c] *Pseudomonas aeruginosa* bacterial meningitis (n = 1), *Escherichia coli* UTI and bacteremia (n = 1)

[d] *Streptococcus pneumoniae* bacterial meningitis and bacteremia (n = 1), *Haemophilus influenzae* bacterial meningitis and sepsis (n = 1), *Streptococcus mitis* bacteremia (n = 1)

## Differential expression of biomarkers and combinational scores

Biomarker levels were assessed in children with FWS and in control patients for the ImmunoXpert and its components (Fig 2). CRP, PCT, WBC, ANC, Labscore and ImmunoXpert values were significantly higher in children across both age groups with bacterial infections compared to children with viral infections or control patients. TRAIL was significantly higher in patients with viral infections, but lower in children with bacterial infections than in control patients. IP-10 was higher in patients with viral infections and in patients with bacterial infections than in controls. CRP, TRAIL, and IP-10 were expressed differently in children < 90 and ≥ 90 days (S1 Fig).

## Diagnostic performance of biomarkers and combinatorial scores

The discriminatory potential of the individual biomarkers and combinatorial scores was assessed using Receiver Operating Characteristic (ROC) analysis (Fig 3). Labscore, ImmunoXpert and CRP attained the highest area under the curve (AUC) values across both age groups (Fig 3). This finding was consistent across the unanimous reference standard subgroup although AUC values were generally lower in the former (S2 Table). The ImmunoXpert interestingly showed a statistically significant difference in its AUC value when compared to that of PCT, ANC, WBC, bands, IP-10, and TRAIL, but not when tested against the Labscore and CRP (Fig 3). Data for the unanimous reference standard subgroup show an identical trend (S2 Table).

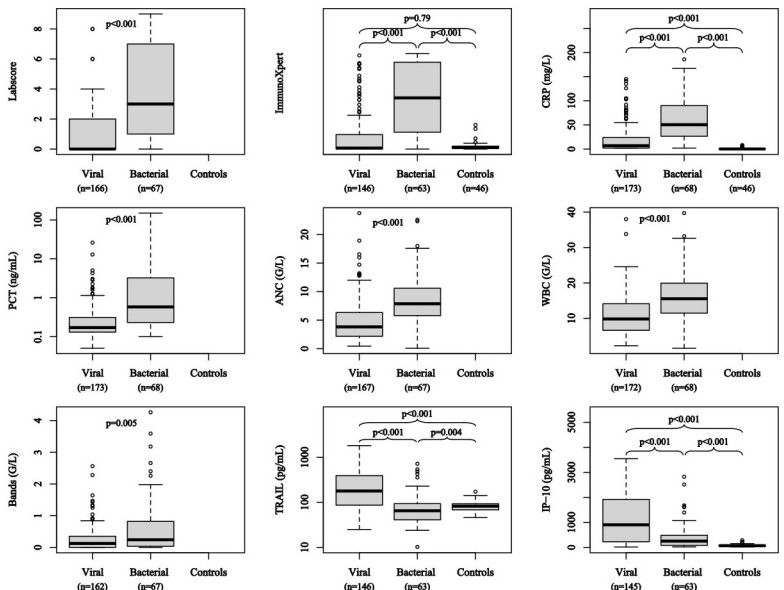

**Fig 2. Distribution of biomarker concentrations according to the final diagnosis.** Distribution of biomarker concentrations in patients with fever without source according to the final diagnosis (viral or bacterial infection) set by expert panel adjudication (and in control patients for ImmunoXpert and its components). Box plots represent median value and interquartile range of the biomarker under study in each group. When comparing groups two by two, p value represents the probability that random chance could produce the observed results when the null hypothesis is true (i.e. when biomarker concentration is equivalent between the two groups under study).

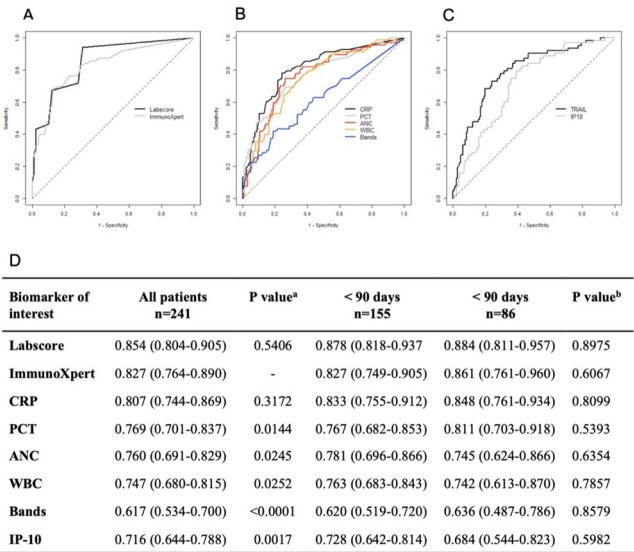

| Biomarker of interest | All patients n=241 | P value[a] | < 90 days n=155 | < 90 days n=86 | P value[b] |
|---|---|---|---|---|---|
| Labscore | 0.854 (0.804-0.905) | 0.5406 | 0.878 (0.818-0.937) | 0.884 (0.811-0.957) | 0.8975 |
| ImmunoXpert | 0.827 (0.764-0.890) | - | 0.827 (0.749-0.905) | 0.861 (0.761-0.960) | 0.6067 |
| CRP | 0.807 (0.744-0.869) | 0.3172 | 0.833 (0.755-0.912) | 0.848 (0.761-0.934) | 0.8099 |
| PCT | 0.769 (0.701-0.837) | 0.0144 | 0.767 (0.682-0.853) | 0.811 (0.703-0.918) | 0.5393 |
| ANC | 0.760 (0.691-0.829) | 0.0245 | 0.781 (0.696-0.866) | 0.745 (0.624-0.866) | 0.6354 |
| WBC | 0.747 (0.680-0.815) | 0.0252 | 0.763 (0.683-0.843) | 0.742 (0.613-0.870) | 0.7857 |
| Bands | 0.617 (0.534-0.700) | <0.0001 | 0.620 (0.519-0.720) | 0.636 (0.487-0.786) | 0.8579 |
| IP-10 | 0.716 (0.644-0.788) | 0.0017 | 0.728 (0.642-0.814) | 0.684 (0.544-0.823) | 0.5982 |

**Fig 3. ROC curves and AUC for the ability of main parameters and biomarkers to detect bacterial infections according to expert panel diagnoses (majority reference standard).** (A) Biological scores: Labscore and ImmunoXpert, (B) Biomarkers for bacterial infections: CRP, PCT, ANC, WBC, and bands, (C) Biomarkers for viral infections: TRAIL, IP-10 as an inverse relationship. The dashed lines represent a non-informative diagnostic biomarker. The specificity at a cut-off is the probability that a viral infection diagnosed by the expert panel is correctly identified by the biomarker and the sensitivity is the probability that a bacterial infection diagnosed by the expert panel is correctly identified by the biomarker. For Labscore, ImmunoXpert, CRP, PCT, ANC, WBC, and bands (resp. IP10 and TRAIL), the area under the ROC curve (D) is an estimate of the probability (95% CI). [a] P value for testing the hypothesis that the area under the ROC curve is the same with the ImmunoXpert (all patients), [b] P value for testing the hypothesis that the area under the ROC curve is equal in patients less than 90 days and in patients more than 90 days.

Pre-defined single cut-offs were applied to biomarkers and dual cut-offs to scores to assess their sensitivity and specificity for bacterial infection across age groups (Table 2). Labscore and ImmunoXpert outperformed single biomarkers with higher sensitivity and/or specificity and exhibited comparable performance to one another. No adverse event was recorded from performing the tests.

Of the 142 patients (68% of total patients) identified with an ImmunoXpert value < 35 at low risk for bacterial infection, 19 of them (13%) were classified as bacterial infections according to the diagnosis set by the panel expert and 10 (7%) of them according to microbiological definitions. Of the 19 patients classified as bacterial infections by the expert panel, 8 patients were diagnosed with fever without source, 9 with urinary tract infection, 1 with otitis media, and 1 with both enteroviral meningitis and *E. Coli* urinary tract infection.

When microbiological diagnosis served as reference standard, trends in AUC were similar when comparing the scores and biomarkers under study, although each individual biological score or biomarker showed a slightly lower AUC value (S3 Table). This trend was similar in patients more and less than 90 days of age.

## Specific performances in the detection of IBI

Five patients were diagnosed with IBI: three with bacterial meningitis (*S. pneumoniae*, *H. influenzae* and *P. aeruginosa*) and/or two with bacteremia (*E. coli* and *S. mitis*). They would all have been detected using an ImmunoXpert ≥ 35, a Labscore threshold of 1 or 3, or PCT ≥ 0.5 ng/mL, whereas WBC ≥ 15 G/L or CRP ≥ 40 mg/L would have detected respectively only two and three of these infections (S4 Table). Out of the four children with IBI tested for the ImmunoXpert, three attained scores ≥ 90. One of them would not have been detected if a threshold of 65 had been applied. However, the ImmunoXpert performed in this 28-day-old patient was exploratory since age was out of the recommended age range of the assay. Among patients with ImmunoXpert values < 35, none of them suffered from an underlying IBI.

## Discussion

The present study assessed the differential expression of individual biomarkers and combinatorial scores and compared their diagnostic performances to reliably predict bacterial infection in children with FWS. For children aged under and older than 90 days, the ImmunoXpert and the Labscore exhibited the highest performance. These findings were consistent whether the reference standard was based on majority adjudication (etiology assigned to every eligible case), unanimous reference standard subgroup (indeterminate cases excluded) or microbiological diagnosis.

The results support that the ImmunoXpert is a diagnostically accurate tool in the evaluation of patients with FWS. This assay had been derived from a population of febrile children aged more than 3 months old and adults with various underlying etiologies [9]. Replacing the classically used CRP with this combination assay improved the diagnostic accuracy to predict pneumonia or other SBIs [25]. However, it had never been tested specifically in the subgroup of pediatric patients suffering from FWS. Data in children less than 3 months were also limited [16]. Previous evaluation of the diagnostic performances of the ImmunoXpert in children had shown 93.8% sensitivity and 89.8% specificity for the diagnosis of bacterial infection. However, this validation study assessed a truncated sample of febrile children, after exclusion of patients with indeterminate diagnoses or those with equivocal ImmunoXpert scores since equivocal score should not be used in decision making [11]. To stay as close as possible to real conditions, patients with indeterminate diagnosis, who represented a non-negligible proportion of patients (28%) and patients with equivocal ImmunoXpert scores were not excluded in our

**Table 2. Diagnostic performances of individual biomarkers and biological scores compared to expert panel adjudication and to microbiological diagnosis as reference standards.**

| | Microbiological diagnosis | | Expert panel diagnosis | | | | | |
| --- | --- | --- | --- | --- | --- | --- | --- | --- |
| | All patients | | All patients | | Age < 90 days | | Age ≥ 90 days | |
| | n/N; Se (%) (95% CI) | n/N, Sp (%) (95% CI) | n/N, Se (%) (95% CI) | n/N, Sp (%) (95% CI) | n/N, Se (%) (95% CI) | n/N, Sp (%) (95% CI) | n/N, Se (%) (95% CI) | n/N, Sp (%) (95% CI) |
| **ImmunoXpert** | | | | | | | | |
| ≥1 | 37/38; 97.4 (86.2–99.9) | 68/171; 39.8 (32.4–47.5) | 58/63; 92.1 (82.4–97.4) | 64/146; 43.8 (35.6–52.3) | 37/41; 90.2 (76.9–97.3) | 50/93; 53.8 (43.1–64.2) | 21/22; 95.5 (77.2–99.9) | 14/53; 26.4 (15.3–40.3) |
| ≥10 | 32/38; 84.2 (68.7–94.0) | 110/171; 64.3 (56.7–71.5) | 50/63; 79.4 (67.3–88.5) | 103/146; 70.5 (62.4–77.8) | 31/41; 75.6 (59.7–87.6) | 72/93; 77.4 (67.6–85.4) | 19/22; 86.4 (65.1–97.1) | 31/53; 58.5 (44.1–71.9) |
| ≥35 | 28/38; 73.7 (56.9–86.6) | 132/171; 77.2 (70.2–83.3) | 44/63; 69.8 (57.0–80.8) | 123/146; 84.2 (77.3–89.7) | 27/41; 65.9 (49.4–79.9) | 83/93; 89.2 (81.1–94.7) | 17/22; 77.3 (54.6–92.2) | 40/53; 75.5 (61.7–86.2) |
| >65 | 13/38; 34.2 (19.6–51.4) | 149/171; 87.1 (81.2–91.8) | 25/63; 39.7 (27.6–52.8) | 136/146; 93.2 (87.8–96.7) | 12/41; 29.3 (16.1–45.5) | 89/93; 95.7 (89.4–98.8) | 13/22; 59.1 (36.4–79.3) | 47/53; 88.7 (77.0–95.7) |
| **Labscor** | | | | | | | | |
| ≥1 | 41/41; 100 (91.4–100) | 118/192; 61.5 (54.2–8.4) | 63/67; 94.0 (85.4–98.3) | 114/166; 68.7 (61.0–75.6) | 40/44; 90.9 (78.3–97.5) | 87/108; 80.6 (71.8–87.5) | 23/23; 100 (85.2–100) | 27/58; 46.6 (33.3–60.1) |
| ≥3 | 32/41; 78.0 (62.4–89.4) | 159/192; 82.8 (76.7–87.9) | 45/67; 67.2 (54.6–78.2) | 146/166; 88.0 (82.0–92.5) | 24/44; 54.5 (38.8–69.6) | 104/108; 96.3 (90.8–99.0) | 21/23; 91.3 (72.0–98.9) | 42/58; 72.4 (59.1–83.3) |
| **PCT** | | | | | | | | |
| ≥0.5 ng/mL | 23/41; 56.1 (39.7–71.5) | 157/200; 78.5 (72.2–84.0) | 36/68; 52.9 (40.4–65.2) | 143/173; 82.7 (76.2–88.0) | 19/45; 42.2 (27.7–57.8) | 98/110; 89.1 (81.7–94.2) | 17/23; 73.9 (51.6–89.8) | 45/63; 71.4 (58.7–82.1) |
| **CRP** | | | | | | | | |
| ≥40 mg/L | 26/41; 63.4 (46.9–77.9) | 153/200; 76.5 (70.0–82.2) | 42/68; 61.8 (49.2–73.3) | 142/173; 82.1 (75.5–87.5) | 21/45; 46.7 (31.7–62.1) | 104/110; 94.5 (88.5–98.0) | 21/23; 91.3 (72.0–98.9) | 38/63; 60.3 (47.2–72.4) |
| **WBC** | | | | | | | | |
| ≥15 G/L | 21/41; 51.2 (35.1–67.1) | 148/199; 74.4 (67.7–80.3) | 36/68; 52.9 (40.4–65.2) | 136/172; 79.1 (72.2–84.9) | 21/45; 46.7 (31.7–62.1) | 92/109; 84.4 (76.2–90.6) | 15/23; 65.2 (42.7–83.6) | 44/63; 69.8 (57.0–80.8) |
| **Bands** | | | | | | | | |
| ≥1.5 G/L | 7/40; 17.5 (7.3–32.8) | 184/189; 97.4 (93.9–99.1) | 9/67; 13.4 (6.3–24.0) | 159/162; 98.1 (94.7–99.6) | 3/44; 6.8 (1.4–18.7) | 103/104; 99.0 (94.8–100.0) | 6/23; 26.1 (10.2–48.4) | 56/58; 96.6 (88.1–99.6) |
| **ANC** | | | | | | | | |
| ≥10 G/L | 11/40; 27.5 (14.6–43.9) | 165/194; 85.1 (79.2–89.8) | 23/67; 34.2 (23.2–46.9) | 150/167; 89.8 (84.2–94.0) | 11/44; 25.0 (13.2–40.3) | 100/107; 93.5 (87.0–97.3) | 12/23; 52.2 (30.6–73.2) | 50/60; 83.3 (71.5–91.7) |
| **IP-10** | | | | | | | | |
| <500 pg/mL | 30/38; 78.9 (62.7–90.4) | 96/170; 56.5 (48.7–64.0) | 48/63; 76.2 (63.8–86.0) | 89/145; 61.4 (52.9–69.3) | 30/41; 73.2 (57.1–85.8) | 56/92; 60.9 (50.1–70.9) | 18/22; 81.8 (59.7–94.8) | 33/53; 62.3 (47.9–75.2) |
| <2000 pg/mL | 37/38; 97.4 (86.2–99.9) | 35/170; 20.6 (14.8–27.5) | 61/63; 96.8 (89.0–99.6) | 34/145; 23.4 (16.8–31.2) | 40/41; 97.6 (87.1–99.9) | 29/92; 31.5 (22.2–42.0) | 21/22; 95.5 (77.2–99.9) | 5/53; 9.4 (3.1–20.7) |
| **TRAIL** | | | | | | | | |

*(Continued)*

**Table 2.** (Continued)

| | Microbiological diagnosis | | Expert panel diagnosis | | | | | |
|---|---|---|---|---|---|---|---|---|
| | All patients | | All patients | | Age < 90 days | | Age ≥ 90 days | |
| | n/N; Se (%) (95% CI) | n/N, Sp (%) (95% CI) | n/N, Se (%) (95% CI) | n/N, Sp (%) (95% CI) | n/N, Se (%) (95% CI) | n/N, Sp (%) (95% CI) | n/N, Se (%) (95% CI) | n/N, Sp (%) (95% CI) |
| <70 pg/mL | 21/38; 55.3 (38.3–71.4) | 131/171; 76.6 (69.5–82.7) | 36/63; 57.1 (44.0–69.5) | 121/146; 82.9 (75.8–88.6) | 22/41; 53.7 (37.4–69.3) | 80/93; 86.0 (77.3–92.3) | 14/22; 63.6 (40.7–82.8) | 41/53; 77.4 (63.8–87.7) |
| <400 pg/mL | 37/38; 97.4 (86.2–99.9) | 39/171; 22.8 (16.7–29.8) | 59/63; 93.7 (84.5–98.2) | 36/146; 24.7 (17.9–32.5) | 38/41; 92.7 (80.1–98.5) | 28/93; 30.1 (21.0–40.5) | 21/22; 95.5 (77.2–99.9) | 8/53; 15.1 (6.7–27.6) |

The counts reported in the columns « Sensitivity » (Se) are the number of bacterial infections as diagnosed according to expert panel (N) and the number of these infections which are correctly identified by the corresponding biomarker (n). The sensitivity is the proportion of bacterial infections correctly identified by a biomarker with the specified cut-off. The counts reported in the columns « Specificity » (Sp) are the number of viral infections as diagnosed according to expert panel (N) and the number of these infections which are correctly identified by the biomarker (n). The specificity is the proportion of viral infections correctly identified by markers with the specified cut-off. Total number of patients suffering from bacterial or viral etiology may differ when some of the parameters could not be tested because of insufficient blood sample or technical problem.

study. Findings support that the ImmunoXpert sensitivity is in line with previous studies in the ≥3 months indication for use population (77% for any score ≥35 and 85% in the subgroup of patients evaluated against unanimous reference standard) [11, 16]. The lower-than-expected specificity could be explained by patients with indeterminate assay results. It is however more important to rule out than in any severe condition. Although outside the indication for use population, the AUC for the ImmunoXpert in patients less than 3 months is comparable to that of older children. It is somewhat difficult to contextualize results on the ImmunoXpert in pediatric patients with FWS with the existing literature since this is the first study to assess the assay in this context. The assay has been derived from a population of febrile children aged more than 3 months old and adults with various underlying etiologies. It had never been tested specifically in the subgroup of pediatric patients suffering from FWS only. Data in patients less than 3 months with a definite infection were also limited. TRAIL, IP-10, and CRP exhibit different responses to infection in children under and over 90 days, prompting the need for further studies focusing on the derivation of a new algorithm intended for this specific population. In the present population of children presenting with FWS, the Labscore demonstrated the highest AUC for the diagnosis of bacterial infection, no matter the age group, showing similar trends in patients < and ≥ 90 days old. A Labscore ≥ 3, originally described as the recommended cut-off point, permitted the detection of 100% of patients with underlying IBI. However, a Lab-score ≥ 3 demonstrated 67% sensitivity, 88% specificity and 87% negative predictive value (NPV) to reliably predict SBI and/or IBI in febrile children when tested against the panel expert reference standard, with better discriminative ability in patients aged 90 days old and more. Better results were achieved when tested against microbiological reference standard, with 78% sensitivity, 83% specificity and 95% NPV. The diagnostic accuracy of this biomarker to detect underlying bacterial infection in febrile children with FWS is lower than in the original derivation and validation retrospective studies [6, 7] but in concordance with the results from prospective studies [8, 26, 27]. As in previous studies on FWS, the proportion of patients with UTIs is high. These patients were classified as bacterial infections by the experts.

In the lack of kidney dimercaptosuccinic acid (DMSA) scan to confirm or exclude true renal involvement, asymptomatic bacteriuria or febrile cystitis could account for false negative cases with no or only mild increase in inflammatory biomarkers.

Most of the literature on the value of biomarkers in febrile children less than 3 months of age has focused on WBC or ANC count in the past, showing no added value of this parameter in detecting bacterial infections [28–30]. Our results confirm insufficient sensitivity of these parameters.

In the present study, PCT shows better diagnostic performances in patients older than 90 days than in younger patients. The superiority of PCT and CRP over WBC and ANC in the identification of IBIs in children was recently published [31]. The results in our study do not support the previously described superiority of PCT over CRP in either age group [32]. This could be explained by different PCT thresholds and reference standard outcomes.

Our study has limitations. First, test accuracy may suffer from biased estimate of sensitivity and specificity and reduced predictive values when forcing expert panels to make a dichotomous decision on target disease classification in the presence of uncertainty [33]. Second, the delay between onset of fever and presentation, which could modify any biological value, was highly variable. Last, CRP and PCT which are included in the biological scores under study, were available to experts assigning the diagnosis. A potential overestimation of the accuracy of tests including any of the above-mentioned biomarkers may have occurred. Interestingly, expert panel diagnosis and microbiological diagnosis show equivalent diagnostic performances with even better accuracy when pure microbiological definitions were applied, showing that this effect is probably minimal.

## Conclusion

The Labscore and the ImmunoXpert show better diagnostic performances for safely detecting bacterial infection in children with FWS, compared to individual biomarkers such as CRP, PCT, WBC or ANC. Better performances were achieved in children more than 90 days old than in younger patients. Evaluation of combined biological scores and biomarkers in various age groups should encourage the development of decision-making rules based on the most appropriate cut-offs permitting to rule out SBI and IBI. Furthermore, implementation of appropriate point-of-care devices and medico-economic impacts of these approaches should be evaluated, notably in their usefulness in safely decreasing unnecessary antibiotic prescription and admission rates and improving the care of children with FWS.

## Supporting information

**S1 Fig. Distribution of biomarker concentrations according to the final diagnosis (unanimous diagnosis reference standard).** Distribution of biomarker concentrations in patients with FWS according to the final diagnosis (viral or bacterial infection) set by expert panel adjudication (and in control patients for ImmunoXpert and its components) in the subgroup analyzed under the unanimous reference standard (exclusion of children showing non concordant etiology labels assigned by the 3 experts, with an indeterminate etiology or showing equivocal ImmunoXpert score $35 \leq$ ImmunoXpert score $\leq 65$).
(TIF)

**S1 Table. STARD checklist.**
(DOCX)

**S2 Table. Diagnostic performance of individual biomarkers (A) and combinatorial scores (B) compared to unanimous reference standard.** [a] The ImmunoXpert is intended for

patients $\geq$ 90 days. The analysis of the ImmunoXpert performance in patients $<$ 90 days is exploratory.
(DOCX)

**S3 Table. Areas under the ROC curves (95% CI) for individual biomarkers and combinatorial scores for the detection of bacterial infection according to microbiological diagnosis.** [a] P value for testing the hypothesis that the area under the ROC curve is equal in patients less than 90 days and in patients more than 90 days. For Labscore, ImmunoXpert, CRP, PCT, ANC, WBC, and bands (resp. IP10 and TRAIL), the area under the ROC curve is an estimate of the probability.
(DOCX)

**S4 Table. Description of patients with microbiologically confirmed invasive bacterial infection.** * Missing data due to insufficient sample.
(DOCX)

**S1 File. Study protocol.**
(DOCX)

## Acknowledgments

We are very grateful to all participating parents and infants, to co-investigators, research nurses, notably Mrs Florence Hugon, and participating clinicians.

## Author Contributions

**Conceptualization:** Laurence Lacroix, Arnaud L'Huillier, Christophe Combescure, Annick Galetto-Lacour, Alain Gervaix.

**Data curation:** Laurence Lacroix, Sebastien Papis, Chiara Mardegan, Fanny Luterbacher, Arnaud L'Huillier, Annick Galetto-Lacour.

**Formal analysis:** Laurence Lacroix, Sebastien Papis, Arnaud L'Huillier, Cyril Sahyoun, Kristina Keitel, Niv Mastboim, Liat Etshtein, Liran Shani, Einav Simon, Eran Barash, Roy Navon, Tanya M. Gottlieb, Kfir Oved, Eran Eden, Christophe Combescure, Annick Galetto-Lacour, Alain Gervaix.

**Funding acquisition:** Arnaud L'Huillier, Annick Galetto-Lacour, Alain Gervaix.

**Investigation:** Laurence Lacroix, Sebastien Papis, Chiara Mardegan, Fanny Luterbacher, Arnaud L'Huillier, Cyril Sahyoun, Kristina Keitel, Niv Mastboim, Liat Etshtein, Liran Shani, Einav Simon, Eran Barash, Roy Navon, Tanya M. Gottlieb, Kfir Oved, Eran Eden, Annick Galetto-Lacour, Alain Gervaix.

**Methodology:** Laurence Lacroix, Arnaud L'Huillier, Christophe Combescure, Annick Galetto-Lacour, Alain Gervaix.

**Project administration:** Laurence Lacroix, Arnaud L'Huillier, Annick Galetto-Lacour, Alain Gervaix.

**Resources:** Laurence Lacroix, Arnaud L'Huillier, Annick Galetto-Lacour, Alain Gervaix.

**Supervision:** Laurence Lacroix, Arnaud L'Huillier, Annick Galetto-Lacour, Alain Gervaix.

**Validation:** Laurence Lacroix, Arnaud L'Huillier, Annick Galetto-Lacour, Alain Gervaix.

**Visualization:** Laurence Lacroix, Arnaud L'Huillier, Annick Galetto-Lacour, Alain Gervaix.

**Writing – original draft:** Laurence Lacroix, Christophe Combescure, Annick Galetto-Lacour, Alain Gervaix.

**Writing – review & editing:** Laurence Lacroix, Sebastien Papis, Chiara Mardegan, Fanny Luterbacher, Arnaud L'Huillier, Cyril Sahyoun, Kristina Keitel, Niv Mastboim, Liat Etshtein, Liran Shani, Einav Simon, Eran Barash, Roy Navon, Tanya M. Gottlieb, Kfir Oved, Eran Eden, Christophe Combescure, Annick Galetto-Lacour, Alain Gervaix.

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
