## [Decision Letter · Decision Letter 0]

6 Sep 2023

PONE-D-23-17053Host biomarkers and combinatorial scores for the detection of serious and invasive bacterial infection in pediatric patients with fever without sourcePLOS ONE

Dear Dr. Lacroix,

Thank you for submitting your manuscript to PLOS ONE. After careful consideration, we feel that it has merit but does not fully meet PLOS ONE’s publication criteria as it currently stands. Therefore, we invite you to submit a revised version of the manuscript that addresses the points raised during the review process.

We look forward to receiving your revised manuscript.

Kind regards,

Novel N. Chegou, Ph.D

Academic Editor

PLOS ONE

Journal Requirements:

"This was an investigator-driven study. Alain Gervaix received financial support from MeMed to support the study. Annick Galetto-Lacour and Arnaud L'Huillier received funding to support the study from the Gertrude von Meissner Foundation, the Ernst and Lucie Schmidheiny Foundation and the University Hospitals of Geneva's Research and Development Project Grant."

3. Thank you for providing the following Competing Interests Statement:  

"The funders and study sponsors had no role in the design and conduct of the study, collection, management, analysis and interpretation of clinical data, preparation for publication and final approval of the manuscript. The first version of the manuscript was written by Laurence Lacroix. The funder (MeMed) conducted the ImmunoXpert assays and participated in reviewing the manuscript with precious scientific input. Niv Mastboim, Kfir Oved, Tanya Gottlieb, Liran Shani, Liat Etshtein, Einav Simon, Eran Eden, Eran Barash and Roy Navon are/were employees of MeMed; the other authors have indicated they have no financial relationships relevant to this article to disclose."

We note that one or more of the authors is affiliated with the funding organization, indicating the funder may have had some role in the design, data collection, analysis or preparation of your manuscript for publication; in other words, the funder played an indirect role through the participation of the co-authors. 

If the funding organization did not play a role in the study design, data collection and analysis, decision to publish, or preparation of the manuscript and only provided financial support in the form of authors' salaries and/or research materials, please review your statements relating to the author contributions, and ensure you have specifically and accurately indicated the role(s) that these authors had in your study in the Author Contributions section of the online submission form. Please make any necessary amendments directly within this section of the online submission form.  Please also update your Funding Statement to include the following statement: “The funder provided support in the form of salaries for authors [insert relevant initials], but did not have any additional role in the study design, data collection and analysis, decision to publish, or preparation of the manuscript. The specific roles of these authors are articulated in the ‘author contributions’ section.” 

If the funding organization did have an additional role, please state and explain that role within your Funding Statement. 

Please also provide an updated Competing Interests Statement declaring this commercial affiliation along with any other relevant declarations relating to employment, consultancy, patents, products in development, or marketed products, etc.  

Within your Competing Interests Statement, please confirm that this commercial affiliation does not alter your adherence to all PLOS ONE policies on sharing data and materials by including the following statement: ""This does not alter our adherence to  PLOS ONE policies on sharing data and materials.” (as detailed online in our guide for authors http://journals.plos.org/plosone/s/competing-interests). If this adherence statement is not accurate and  there are restrictions on sharing of data and/or materials, please state these. Please note that we cannot proceed with consideration of your article until this information has been declared.

6. Please note that in order to use the direct billing option the corresponding author must be affiliated with the chosen institute. Please either amend your manuscript to change the affiliation or corresponding author, or email us at plosone@plos.org with a request to remove this option.

7. We note that you have included the phrase “data not shown” in your manuscript. Unfortunately, this does not meet our data sharing requirements. PLOS does not permit references to inaccessible data. We require that authors provide all relevant data within the paper, Supporting Information files, or in an acceptable, public repository. Please add a citation to support this phrase or upload the data that corresponds with these findings to a stable repository (such as Figshare or Dryad) and provide and URLs, DOIs, or accession numbers that may be used to access these data. Or, if the data are not a core part of the research being presented in your study, we ask that you remove the phrase that refers to these data.

8. Please include your full ethics statement in the ‘Methods’ section of your manuscript file. In your statement, please include the full name of the IRB or ethics committee who approved or waived your study, as well as whether or not you obtained informed written or verbal consent. If consent was waived for your study, please include this information in your statement as well. 

Reviewers' comments:

Reviewer's Responses to Questions

**Comments to the Author**

1. Is the manuscript technically sound, and do the data support the conclusions?

Reviewer #1: Yes

2. Has the statistical analysis been performed appropriately and rigorously? 

Reviewer #1: Yes

3. Have the authors made all data underlying the findings in their manuscript fully available?

Reviewer #1: Yes

4. Is the manuscript presented in an intelligible fashion and written in standard English?

Reviewer #1: Yes

5. Review Comments to the Author

Reviewer #1: This is an observational diagnostic accuracy stuy by Lacroix and colleagues from Switzerland, who report here findings of a substudy on children below the age of 3 years, with a particular focus on young infants in the first three months of life. The study methodology and reporting seem valid, the findings are important due to the very relevant patient cohort. I have some major comments that however should be addressed:

Major comments

- For readers who may not be familiar with the labscore, I encourage the authors to explain in more detail how the labscore is derived, what cutoffs it has etc.

- Please explain why Canadian children were used as healthy controls for a study on Swiss patients

- Line 247: The data for the unanimous reference standard subgroup should be shown in the supplement.

- Explain the asterisk in Table S4 for patient number 245 in the column Immunoexpert.

- The false negatives are problematic. The authors should elaborate on the failure of the test in these cases, which were bacteremia or meningitis cases, and also discuss and contextualize with previous studies and their false negatives.

- The references seem in part to be older, in light of newer studies on the studied marker, the authors should cite and discuss findings from other studies, e.g. this systematic review: PMID 34015531; a large study on a broad pediatric cohort: PMID 34768022; and a very recent study on older patients: PMID 37270059.

6. PLOS authors have the option to publish the peer review history of their article (what does this mean?). If published, this will include your full peer review and any attached files.

Reviewer #1: No

---

## [Author Response · Author response to Decision Letter 0]

11 Oct 2023

Dear Editor, 

We are grateful for the revision of our manuscript entitled: “Host biomarkers and combinatorial scores for the detection of serious and invasive bacterial infection in pediatric patients with fever without source” and for your comments and suggestions. You will find below a detailed point by point response to each point raised during the review process to correspond to the journal requirements and to address the comments of both the academic editor and the reviewers.

Response to the Academic Editor and Journal requirements: 

1. PLOS ONE's style requirements: our manuscript has been checked so that it meets PLOS ONE's style requirements, including those for file naming. 

2. Funding of the study: there was no additional external funding other than those already mentioned in the original funding statement. As you suggested, our Funding Statement should be updated as follows: 

“This was an investigator-driven study. Alain Gervaix received financial support from MeMed to support the study. The funder provided support in the form of salaries for authors NM, LE, LS, ES, EB, RN, TG, KO, EE. The specific roles of these authors are articulated in the ‘author contributions’ section. The funders and study sponsors had no role in the design and conduct of the study, and in interpretation of clinical data. The first version of the manuscript was written by Laurence Lacroix. The funder (MeMed) participated in data collection and analysis (ImmunoXpert assays) and in reviewing the manuscript. Annick Galetto-Lacour and Arnaud L'Huillier received funding to support the study from the Gertrude von Meissner Foundation, the Ernst and Lucie Schmidheiny Foundation and the University Hospitals of Geneva's Research and Development Project Grant. There was no additional external funding received for this study.” We would be grateful if you change the online submission form on our behalf.

3. Competing Interests Statement. Thank you for your comment on author affiliation. We had previously mentioned that some of the authors are employees of MeMed, one of the funding organizations. Our statements relating to the author contributions have been revised in the Author Contributions section of the online submission form. The amended statement of competing interests now explicitly states: “The financial support received by AG from MeMed as well as the commercial affiliation of authors NM, LE, LS, ES, EB, RN, TG, KO, EE with MeMed does not alter our adherence to all PLOS ONE policies on sharing data and materials. Other authors had no relationship relating to employment, consultancy, patents, products in development, marketed products, etc. with any of the funding organizations.”

4 & 5. Data Availability statement. We had previously stated that data were available “upon request”. We now confirm that all data underlying the findings in our study will be freely available from a public repository (Dryad) as soon as the manuscript is accepted for publication (https://doi.org/10.5061/dryad.gxd2547s8). This information has been updated in our Data Availability statement. However, as suggested, before before uploading the data and abstract and making them public, we will wait for your confirmation that the manuscript is accepted for publication. 

6. Author affiliation. The corresponding author (LL) is affiliated with the institute chosen for direct billing option (University of Geneva, Geneva, Switzerland) (page 1 lines 6 and 13).

7. The term “data not shown” was not exact. We were meaning “results not shown in the main manuscript file”. However, this assertion refers to an already mentioned table in the supporting information file (S2 Table) that shows all data underlying this comment. The phrase “data not shown” has been replaced by the indication for the corresponding table in the supporting information file (S2 Table) (page 12, line 280).

8. Ethics statement. The full name of ethics committee (Cantonal Committee on Ethics in Scientific Research of Geneva, CCER) who approved our study has been added (page 6 line 146). The need for obtaining written informed consent before inclusion has been added in the revised version of the manuscript (page 6 line 148). 

9. Reference list. The reference list has been updated to address the comments from the Editor (details on the Labscore, ref 7-8) and from Reviewer # 1, see below (ref 13-14).

Response to the Reviewers: 

- Comment from the Reviewer: For readers who may not be familiar with the labscore, I encourage the authors to explain in more detail how the labscore is derived, what cutoffs it has etc.

Author response: Thank you for your suggestion. Details on the Labscore (derivation and validation of the score), and on the cutoffs and recommendations for use have been added in the revised version of the manuscript (page 3-4 line 70-81).

- Comment from the Reviewer: Please explain why Canadian children were used as healthy controls for a study on Swiss patients.

Author response: Thank you for your comment. One of the goals in our study was the determination of differential biomarker expression between children with fever without source and healthy children. Facing the ethical challenge of performing blood puncture in healthy control children with no fever and no current infection, we decided to use preexisting blood samples to test the relevant biomarkers. We did not find any existing healthy pediatric control samples in Switzerland, but we found an existing healthy Canadian cohort to address this precise objective. This has been added to the revised version of the manuscript (page 6 line 137-143).

- Comment from the Reviewer: Line 247: The data for the unanimous reference standard subgroup should be shown in the supplement.

Author response: As mentioned above in the response to the academic editor comments, the term “data not shown” referred to a table not shown in the main manuscript file (but that was already included in the supplemental online materiel as S2 Table). The phrase “data not shown” has been replaced by the corresponding table in the supporting information file (S2 Table) (page 12, line 280).

- Comment from the Reviewer Explain the asterisk in Table S4 for patient number 245 in the column ImmunoXpert.

Author response: Thank you for this detail that we had missed. The asterisk in Table S4 in the ImmunoXpert column of patient 245 refers to insufficient blood sample to perform the test. This has been added in the Supporting Information section of the revised manuscript (page 22 line 553).

- Comment from the Reviewer: The false negatives are problematic. The authors should elaborate on the failure of the test in these cases, which were bacteremia or meningitis cases, and also discuss and contextualize with previous studies and their false negatives. 

Author response: Thank you for your suggestion concerning false negatives cases that are indeed problematic. We had previously elaborated on patients with invasive bacterial infections (IBI). All five of them, included bacteremia or meningitis cases mentioned above, would have been detected using an ImmunoXpert ≥ 35, a Labscore threshold of 1 or 3, or PCT ≥ 0.5 ng/mL. However, WBC ≥ 15 G/L or CRP ≥ 40 mg/L would have detected respectively only two and three of these infections (page 15 line 333-341). False negative cases for the ImmunoXpert were patients with a low ImmunoXpert score (<35) but classified as bacterial infections by the panel of experts. As stated in the text (page 15 line 322-327), these cases were mainly patients with a final diagnosis of fever without source, or urinary tract infections. In our study, no DMSA scan was performed to differentiate UTI with or without kidney involvement. Patients with UTI diagnosis and low (<35) ImmunoXpert scores could represent the proportion of patients with cystitis, which is not a serious bacterial infection, but which can present with low grade fever in up to 30% of cases according to the literature, as it was already stated in the manuscript (page 18 line 394-397). The ImmunoXpert assay has been derived from a population of febrile children aged more than 3 months old and adults with various underlying etiologies. It has never been tested specifically in the subgroup of pediatric patients suffering from FWS and data in children less than 3 months were also limited in patients with a definite infection. It is somewhat difficult to contextualize results on the ImmunoXpert in pediatric patients with FWS with the existing literature since this is the first study to assess the assay in this context. This has been added in the discussion section (page 17 line 373-379). 

- Comment from the Reviewer: The references seem in part to be older, in light of newer studies on the studied marker, the authors should cite and discuss findings from other studies, e.g. this systematic review: PMID 34015531; a large study on a broad pediatric cohort: PMID 34768022; and a very recent study on older patients: PMID 37270059.

Author response: Thank you for the suggested references. The systematic review that is mentioned focuses on biomarkers in both adult and pediatric patients with respiratory tract infection, which is not the scope of our study, but it has now been cited as a reference (ref 14, PMID 34015531). The study on the TRAIL/IP-10/CRP signature in a broad pediatric cohort refers to children > 90 days of age with either respiratory tract infections or fever without source. It has been cited as a reference in the introduction section (ref 13, PMID 34768022). The third reference mentioned (PMID 37270059) has not been cited since it refers to febrile adult patients only (median age 56 years old) with suspected LRTI (and not with fever without source).

We are grateful to the Reviewers and the Editor for the time spent in revising our manuscript and for their precious input and for considering this article for publication. 

Sincerely,

Dr Laurence Lacroix-Ducardonnoy, MD

Pediatric Emergency Department

Children’s Hospital, University Hospitals of Geneva and Geneva University

---

## [Decision Letter · Decision Letter 1]

23 Oct 2023

Host biomarkers and combinatorial scores for the detection of serious and invasive bacterial infection in pediatric patients with fever without source

PONE-D-23-17053R1

Dear Dr. Lacroix

We’re pleased to inform you that your manuscript has been judged scientifically suitable for publication and will be formally accepted for publication once it meets all outstanding technical requirements.

Kind regards,

Novel N Chegou, Ph.D

Academic Editor

PLOS ONE

Additional Editor Comments (optional):

Reviewers' comments:

Reviewer's Responses to Questions

**Comments to the Author**

1. If the authors have adequately addressed your comments raised in a previous round of review and you feel that this manuscript is now acceptable for publication, you may indicate that here to bypass the “Comments to the Author” section, enter your conflict of interest statement in the “Confidential to Editor” section, and submit your "Accept" recommendation.

Reviewer #1: All comments have been addressed

2. Is the manuscript technically sound, and do the data support the conclusions?

Reviewer #1: Yes

3. Has the statistical analysis been performed appropriately and rigorously? 

Reviewer #1: Yes

4. Have the authors made all data underlying the findings in their manuscript fully available?

Reviewer #1: Yes

5. Is the manuscript presented in an intelligible fashion and written in standard English?

Reviewer #1: Yes

6. Review Comments to the Author

Reviewer #1: Thank you for your revisions, all of my comments have been sufficiently addressed now, no further questions.

7. PLOS authors have the option to publish the peer review history of their article (what does this mean?). If published, this will include your full peer review and any attached files.

Reviewer #1: No

---

## [Editor Report · Acceptance letter]

2 Nov 2023

PONE-D-23-17053R1 

Host biomarkers and combinatorial scores for the detection of serious and invasive bacterial infection in pediatric patients with fever without source 

Dear Dr. Lacroix:

I'm pleased to inform you that your manuscript has been deemed suitable for publication in PLOS ONE. Congratulations! Your manuscript is now with our production department. 

Kind regards, 

on behalf of

Prof Novel Njweipi Chegou 

Academic Editor

PLOS ONE